# Proposition of a New Scale for Marginal Bone Loss Prediction Around Dental Implants—A 5-Year Follow-Up of Functional Loaded Implants

**DOI:** 10.3390/diagnostics15243101

**Published:** 2025-12-06

**Authors:** Tomasz Wach, Marcin Kozakiewicz, Adam Michcik, Piotr Hadrowicz, Paulina Pruszyńska, Grzegorz Trybek, Maciej Sikora, Piotr Szymor, Raphael Olszewski

**Affiliations:** 1Department of Maxillofacial Surgery, Medical University of Lodz, 251 Pomorska Str., 92-213 Lodz, Poland; 2Department of Maxillofacial Surgery, Medical University of Gdansk, 80-210 Gdańsk, Poland; 3Department of Otolaryngology, Hospital in Sosnowiec, Zegadłowicza 3, 41-200 Sosnowiec, Poland; 4Department of Oral Surgery, Pomeranian Medical University in Szczecin, 70-111 Szczecin, Poland; 5Department of Maxillofacial Surgery, 4th Military Clinical Hospital in Wroclaw, ul. Rudolfa Weigla 5, 50-981 Wroclaw, Poland; 6National Medical Institute of the Ministry of Interior and Administration, Wołoska 137 Str., 02-507 Warsaw, Poland; 7Department of Maxillofacial Surgery, Hospital of the Ministry of Interior, Wojska Polskiego 51, 25-375 Kielce, Poland; 8Department of Biochemistry and Medical Chemistry, Pomeranian Medical University, Powstańców Wielkopolskich 72, 70-111 Szczecin, Poland; 9Department of Oral and Maxillofacial Surgery, Cliniques Universitaires Saint Luc, UCLouvain, 1200 Brussels, Belgium; 10Oral and Maxillofacial Surgery Research Lab (OMFSLab), NMSK, Institut de Recherche Expérimentale et Clinique (IREC), UCLouvain, 1200 Brussels, Belgium; 11Department of Perioperative Dentistry, L. Rydygiera Collegium Medicum in Bydgoszcz, Nicolaus Copernicus University in Torun, 85-067 Torun, Poland

**Keywords:** texture analyses, intraoral radiograph image, bone lesion, corticalization, dental implants, marginal bone loss, prognosis, success rate

## Abstract

**Background**: Marginal bone loss (MBL) is a condition leading to implant loss and treatment failure. MBL is one of the main complications in dental implantology. The aim of this research is to show the method that can predict bone loss around implants and protect patients from implant loss. **Methods**: A total of 1026 intraoral standardized radiographs of dental implants were included in this study. A total of 2052 peri-implant jawbone image samples were analyzed in MaZda 4.6 software. A new scale was calculated and described. MBL was measured, and groups of patients were compared. **Results:** After 3 months of functional loading, the Corticalization Index (CI) was calculated to be 210.70 ± 149.78 and increased after 60 months to 277.88 ± 198.78. In the 60-month observation, MBL was 0.85 mm ± 1.29 mm, and it has been noted that low MBL is associated with CI lower than 300, high MBL with CI higher than 500, and critical high MBL appeared when CI was higher than 1200. **Conclusions:** Authors created a new scale, and research showed that the specified CI in our scale may predict MBL around dental implants with a five-year forecast horizon. It allows us to implement specific treatment and protect implants from loss.

## 1. Introduction

Dental implants have become a common and widely available solution for tooth loss and esthetic defects in the dentition. The number of patients with at least one implant is increasing. Despite generally promising treatment outcomes, a significant group of patients (estimated at 10–20%) are affected by peri-implantitis. This is a process that leads to bone loss around the implant [1,2]. Peri-implant mucositis, characterized by mucous membrane inflammation without pathological bone loss, versus peri-implantitis, affecting both soft tissues and bone, are the two described processes of peri-implant disease [2,3].

Dental implant complications include infectious issues (such as peri-implantitis leading to bone loss), prosthetic problems (e.g., screw loosening or fractures), occlusal overload, and systemic health factors that may affect implant success. Peri-implant disease is one of the most significant concerns in contemporary implantology, as it can directly result in dental implant loss and treatment failure. Numerous strategies have been employed to prevent marginal bone loss following the functional loading of implants, including new surgical techniques, modifications to implant design, thorough medical assessments before and after surgery, and efforts to achieve optimal prosthetic restoration. Additionally, marginal bone loss may be significantly influenced by patient habits, such as smoking, as well as by medical history, which should always be considered when planning treatment [4,5,6,7,8].

Considering radiological techniques, CBCT (cone-beam computed tomography) has become a widespread technique, commonly used in dental implant surgery. It is possible to evaluate bone volume and its condition [9]. It is also possible to analyze bone levels in dental implants using intraoral radiographs [10,11].

The aim of this study is to present the Corticalization Scale to predict marginal bone loss around dental implants before it appears using the image texture analysis applied to two-dimensional (2D) intra-oral radiographs (IOR).

## 2. Materials and Methods

This study was conducted with the approval of the Institutional Ethics Committee of the Medical University of Lodz (approval numbers: RNN/485/11/KB, date 14 June 2011).

All dental implants were placed by the same surgeon (M.K.) following a standardized protocol:•Surgery was performed under local anesthesia with Articaine and Adrenaline 1:100,000.•A periosteal flap was created, and the implant site was drilled.•The implant was inserted according to manufacturer-recommended protocols.•The healing process proceeded under a closed mucoperiosteal flap.•All inserted implants were two-stage implants. The healing process proceeded under a closed mucoperiosteal flap, with the implant remaining unloaded during this period.•Following an initial healing period of 3 months, the implant was uncovered under local anesthesia using Articaine with Adrenaline 1:100,000.•Standard healing abutments were placed.•Impressions were taken two weeks later, after soft tissue healing was complete.•Prosthetic restorations were then fabricated and applied.•The patients were observed over a follow-up period of 5 years.

### 2.1. Inclusion Criteria

18 years old or older patients.Bleeding on probing <20%.Good oral hygiene.Gingival pocket depth 3 mm or less. oTwo-dimensional radiographs taken during routine checks and regular follow-ups.Blood tests considered ion and hormone levels: oTSH (normal range 0.23–4.0 µU/mL).oPTH (normal range 10 to 60 pg/mL).oGlycated hemoglobin (normal range < 5%).oIons Ca^2+^ (normal range 9–11 mg/dL).oVitamin D3 (normal range 31–50 ng/mL).Bone mineral density was evaluated by spine densitometry.

### 2.2. Exclusion Criteria

Absence, low quality, or lack of radiographic images during the observation period.Implant loss within the initial 3-month healing period.Lack of laboratory test results.Radiographic images showing defects upon visual assessment.Poorly controlled internal comorbidities.Presence of other immunodeficiencies.History of radiotherapy.Additional soft tissue and/or bone augmentation procedures.Use of cytostatic drugs in the patient’s medical history.

A total of 1270 implants placed in 820 patients were analyzed in this study, which means that there were 2540 samples of Region of Interest (ROI). Considering inclusion and exclusion criteria, 244 implants were excluded from the research (lack of control images, artifacts on radiographs, implant loss in the first 3 months of healing, and a few analyses with disproportionately high values of texture analyses). Finally, 1026 implants were included, and 2052 ROI samples were analyzed.

### 2.3. Data Acquisition

Standardized intraoral radiographs were taken immediately after the surgery (0 M), 3 months after functional loading (3 M), and after 5 years of observation (60 M). Radiographs were captured using the DIGORA OPTIME radiography system (TYPE DXR-50, SOREDEX, Helsinki, Finland) with the following settings: 7 mA, 70 mV, and 0.1 s. The focus apparatus was provided by Instrumentarium Dental, Tuusula, Finland.

Intra-oral radiographs (IORs) were taken by the same researcher through the 5 years of follow-up. Positioners were used to ensure that images were taken repeatedly. Radiologically recorded bone structure was studied through digital texture analysis using the previously proposed version 1 of the Corticalization Index (CI) [12,13].

A total of 2052 samples were analyzed using MaZda 4.6 freeware software (https://qmazda.p.lodz.pl/) developed by the University of Technology in Lodz [14,15] to assess measures of corticalization in the peri-implant environment of trabecular bone (representing original bone before implant-related alterations) and soft tissue (indicative of marginal bone loss).

Prior to defining the regions of interest (ROIs) and measuring marginal bone loss, all radiographic images were corrected for implant long-axis alignment to standardize the analysis. ROIs were marked at consistent locations on intraoral radiographs taken at different time points: 3 months (3 M) and 60 months (60 M) during the observation period. Subsequent analyses of texture features were conducted to calculate the Corticalization Index (Figure 1).

Marginal bone loss (MBL) was measured on radiological images in the standard way: length measured between the platform of the implant and the bottom of the marginal bone loss (Figure 2).

The Corticalization Index was calculated using second-order features. First, differential entropy was calculated as a measure of the overall scatter of bone structure elements in a radiograph. High values of differential entropy are characteristic of cancellous bone [16].
$$DifEntr=-\sum_{i=1}^{Ng}p_{x-y}\left(i\right)log(p_{x-y}(i))$$ where Σ is the sum; Ng is the number of levels of optical density in the radiograph; I and j are the optical density of pixels with a 5-pixel distance from one another; p is probability; and log is the common logarithm.

Next, the last primary texture feature was calculated:
$$LngREmph=\frac{\sum_{i=1}^{Ng}\sum_{k=1}^{Nr}k^{2}p(i,k)}{\sum_{i=1}^{Ng}\sum_{k=1}^{Nr}p(i,k)}$$ where Σ is the sum; Nr is the number of series of pixels with density level I and length k; Ng is the number of levels for image optical density; Nr is the number of pixels in series; and p is probability [17]. This texture feature describes thick, uniformly dense, radio-opaque bone structures in intra-oral radiograph images [18,19].
$$CI=\frac{LngREmph\cdot Mean\;Optical\;Density}{DifEntr}$$

Statistical analysis included feature distribution evaluation, mean (*t*-test) or median (W-test) comparison, regression analysis, and a one-way analysis of variance or the Kruskal–Wallis test as indicated by non-normal distribution or between-group variance on significant differences in the investigated groups. Differences or relationships were assumed to be statistically significant at *p* < 0.05. Statgraphics Centurion version 18.1.12 (StatPoint Technologies, Warrenton, VA, USA) was used for statistical analyses. All tests were performed to show how marginal bone loss appears depending on the changing Corticalization Index over time.

## 3. Results

We found a statistically significant difference (*p* < 0.05) in the Corticalization Index (CI) between measurements taken after 3 months and those taken after 60 months of observation (Figure 3). A similar significant difference was observed for marginal bone loss (MBL) between the 3-month and 60-month follow-up periods (Figure 4, Table 1).

The study revealed a correlation between the Corticalization Index (CI) and marginal bone loss (MBL) after 5 years of follow-up (Table 2), as well as a correlation between CI measured at 3 and 60 months (Figure 5 and Figure 6).

At 3 months, there was no statistically significant difference in marginal bone loss (MBL) between patients with a Corticalization Index (CI) below 300 (0.2 mm ± 0.77) and those with a CI above 500 (0.27 mm ± 0.87) (Figure 7). However, after 5 years of follow-up, patients with CI below 300 exhibited significantly lower MBL (mean 0.73 mm ± 1.21) compared to patients with CI above 500, who showed higher MBL (mean 1.23 mm ± 0.9), with the difference being statistically significant (*p* < 0.05) (Figure 8).

The researcher analyzed the Corticalization Index (CI) at 3 months in patients whose CI was below 300 and above 500 after 60 months of observation. The study revealed a statistically significant difference (*p* < 0.05) between these groups: patients with CI > 500 had a mean CI of 295.3 ± 170.80 at 3 months, while those with CI < 300 had a mean CI of 178.30 ± 95.87. This finding suggests that adverse changes in marginal bone may begin as early as 3 months after functional loading, which is a crucial observation in this study (Figure 9).

The study also demonstrated that marginal bone loss after 60 months of functional loading strongly depends on the Corticalization Index measured at 3 months after loading. Higher CI values were associated with greater MBL, especially when the CI increased up to 1200 (Figure 10).

Our study showed that, in a group of patients where CI was lower than 300 after 3 months of functional loading, the lower MBL was observed after 5 years (60 months). When CI was higher than 500 after 3 months of functional loading, higher MBL was observed after 5 years (Figure 11).

**Figure 11 diagnostics-15-03101-f011:**
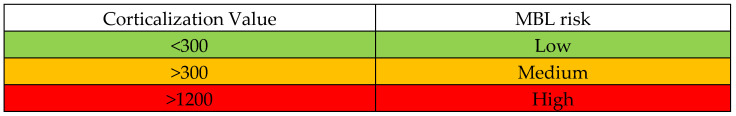
Figure presents the Corticalization Scale and the risk of MBL after 60 months of functional loading, depending on the Corticalization value of the bone near the dental implant 3 months after functional loading. Abbreviations: MBL—marginal bone loss.

## 4. Discussion

### 4.1. Complications and Importance of Follow-Up

Dental implant complications can be broadly categorized into several groups: Infectious complications—including peri-implantitis, which can lead to marginal bone loss and implant failure.

Prosthetic complications—issues related to the implant-supported prosthesis (e.g., screw loosening, fracture of components). Occlusal overload—excessive biting forces on the implant or prosthesis.

Systemic health-related changes—the patient’s general health issues that may affect implant success.

Most of these complications tend to appear within the first five years after implantation [20,21]. Therefore, diligent follow-up in the early years post-surgery is critical for early detection and management of complications. After the surgical procedure, follow-up becomes a crucial aspect of care, consisting of both thorough clinical examinations and appropriate radiological assessments.

### 4.2. Radiological Follow-Up: CBCT Vs. Two-Dimensional Imaging

In implantology follow-up, radiographic evaluation plays a key role. Cone-beam computed tomography (CBCT) is commonly used because it provides a three-dimensional assessment of the bone’s quality and volume around implants. However, despite its high accuracy, CBCT has several disadvantages that can limit its usefulness for detailed bone texture analysis [22,23]:•Artifacts from metal and dense objects: CBCT images often suffer from beam hardening artifacts, seen as dark streaks between metal implants. There is also a photon starvation effect, which generates significant noise and streaks around very dense objects (like implants or metal restorations), obscuring detail.•Voltage-dependent distortion: The apparent size of objects in CBCT images can vary with the X-ray tube voltage. Lower voltage settings may cause objects to appear larger, whereas higher voltages can make them appear smaller. This variability can distort measurements and make reliable texture analysis impossible.

In contrast, two-dimensional radiographic images (such as periapical or panoramic X-rays) do not suffer from the same 3D artifact issues and have proven more suitable for texture analysis of peri-implant bone in this study. A 2D radiograph provides a summation image of the structures, which means the buccal (vestibular) bone plate is not distinctly visible. Despite this limitation, our research showed that texture analysis of standard 2D radiographs taken during follow-up is a useful tool and may serve as a good predictor of potential marginal bone loss around implants.

### 4.3. Prevention: Planning and Patient Selection

One of the most effective ways to avoid implant complications is meticulous prevention through well-planned treatment. Pre-surgical planning often includes radiographic evaluation; using CBCT for treatment planning is a critical first step that helps determine optimal implant placement and angulation [23,24]. Careful patient selection is another key preventive measure. Not every patient is an ideal candidate for dental implants, and in some cases, a traditional prosthetic approach might be more appropriate. Patients with certain systemic conditions are at higher risk for implant complications and implant failure [25,26,27]. These include, for example, patients with osteoporosis who are on bisphosphonate therapy (which can affect bone healing and remodeling); patients with a significant oncological history, especially those who have received bisphosphonates or radiation therapy as part of cancer treatment; individuals with immunodeficiencies or other systemic conditions that impair healing; patients with a history of head/neck radiotherapy affecting the jawbones; and patients on cytostatic (chemotherapy) drugs.

In our study, strict inclusion and exclusion criteria were applied. This meant that patients with poorly controlled systemic diseases, immunodeficiencies, or a history of radiotherapy were generally excluded. Likewise, cases requiring extensive adjunctive procedures (such as significant bone or soft tissue augmentation) or patients with a history of cytostatic drug use were either not included or, if treated with implants, managed with extreme caution. By carefully selecting patients and optimizing their health status before implant therapy, we aimed to minimize the risk of complications. Potentially, using the Corticalization Index (discussed below) at the planning stage provides a way to assess bone status without exposing patients to additional ionizing radiation, offering an early indication of whether the bone is likely to respond favorably to an implant.

### 4.4. Corticalization Index and Marginal Bone Loss

A major finding of this research is the clear correlation between the Radiological Corticalization Index (CI) and the progression of marginal bone loss around implants. The Corticalization Index is a quantitative measure of how the trabecular (spongy) bone around the implant neck changes (corticalizes) over time. We observed that a higher CI measured after the implant has been in function (i.e., after loading) is associated with greater marginal bone loss over the long term. In other words, the more pronounced the corticalization in the peri-implant bone at an early stage, the more bone loss tends to occur by five years of functional loading.

Importantly, by calculating the Corticalization Index at 3 months after functional loading, it may be possible to predict future bone loss around the implant at 60 months (5 years) of follow-up. Based on the 3-month CI values in this study, we propose stratifying patients into risk groups for future marginal bone loss:

Low risk: 3-month CI value < 300 (indicates healthy bone adaptation; minimal risk of significant marginal bone loss by 5 years).

Medium risk: 3-month CI value around or above 500 (indicates an intermediate risk; some caution and closer monitoring are warranted).

High risk: 3-month CI value > 1200 (this is a critical threshold; it indicates a very high risk of future marginal bone loss under the current conditions).

Using this Corticalization Scale, a clinician can estimate the likelihood that marginal bone loss will occur by the 60-month follow-up. If the index after 3 months of loading is low (for example, CI < 300), the risk of future bone loss is low, and routine observation can continue. If the CI is elevated (for example, exceeding 500), it signals a medium or higher risk, and proactive measures should be considered. A value above 1200 is especially alarming, suggesting that significant bone loss is likely if no intervention is performed.

There are a few limitations to using the Corticalization Index method that should be noted:•All follow-up radiographs should be taken with the same radiographic device and settings, to ensure consistency in image quality and grayscale, which affects the CI calculation.•Specialized image analysis software (such as MaZda 4.6) is required to calculate the Corticalization Index from the radiographs, which may not be readily available in all clinical settings.•The CI must be calculated on a defined Region of Interest (ROI) on the radiograph, and this ROI selection must be standardized. In our study, we standardized the ROI based on the method described by Kozakiewicz et al. [10], ensuring that the measurements are reliable and reproducible.•As with any such analysis, operator training is needed to correctly perform texture analysis and interpret the Corticalization Index values.

Despite these limitations, the Corticalization Index proved to be a promising tool in identifying implants at risk of future bone loss, well before such loss becomes radiographically or clinically evident.

### 4.5. Prosthetic Considerations and Early Interventions

After an implant has healed (typically around 3 months post-surgery) and is ready for functional loading, the clinician must choose the optimal prosthetic reconstruction for the missing dentition. The design of the prosthetic components (for example, the emergence profile of the crown or the abutment shape) can influence the health of the surrounding bone. Certain prosthetic choices—such as a poorly contoured emergence profile that encroaches on soft tissue or is difficult to keep clean—may negatively affect marginal bone preservation [28,29,30,31]. Therefore, the period shortly after loading is critical: the first detailed follow-up assessment is recommended about 3 months after the implant has been put into function (with its prosthetic restoration).

Our findings suggest that evaluating the peri-implant bone at this 3-month post-loading visit using the Corticalization Index can help predict if the current prosthetic restoration is likely to lead to future bone loss by 5 years. If a patient’s 3-month CI is in the higher range (for instance, >500, indicating medium or elevated risk), this should alert the dentist that the present restoration might contribute to marginal bone loss over time. In such cases, the dentist should consider adjusting or redesigning the prosthetic restoration—for example, modifying the crown’s emergence profile, improving occlusal contacts, or enhancing oral hygiene access—to mitigate the risk of bone loss. On the other hand, if the CI remains low (<300) at 3 months, it suggests the restoration and implant are in a favorable balance with the bone, and the patient can continue with standard monitoring protocols.

Regular clinical and radiological follow-up should continue beyond the initial 3-month check. By catching unfavorable trends early (such as a rising CI or early signs of bone change), clinicians can intervene—whether by changing prosthetic components, improving occlusion, or instituting other therapies—to preserve the bone and implant support.

### 4.6. Occlusal Overload and Maintenance of Implant Success

During long-term follow-up of implant patients, it is also important to monitor changes in the patient’s dentition and occlusion. Over the years, patients may experience loss of other teeth, advanced tooth wear (abrasion), or parafunctional habits like bruxism. These changes can alter the distribution of biting forces. When teeth are lost or worn down, the overall support for occlusion is reduced, meaning the dental implants (along with any remaining teeth) may have to bear heavier loads with fewer contact points. This scenario—occlusal overload—is a known pathway to accelerated marginal bone loss around implants.

Excessive occlusal forces can cause micro-damage to the bone-implant interface and lead to bone resorption over time. Therefore, at each follow-up visit, the dentist should evaluate the occlusal condition. If signs of occlusal imbalance or overload are present, corrective measures should be taken. This could include restoring missing teeth (to redistribute forces), adjusting the bite (equilibration), or providing a night guard for bruxism. Patients must be informed about the importance of managing these issues; any newly developed deficiencies or dysfunctions in the stomatognathic system should be addressed promptly [32,33].

Occlusal overload can lead to unfavorable bone remodeling and even atrophy of the supporting bone if uncorrected. An early warning of such a problem could be detected through the Corticalization Index as well. For instance, a steadily increasing CI in subsequent checkups might indicate that the bone is becoming denser (sclerotic) in response to stress, which often precedes bone loss. Recognizing this pattern would prompt the clinician to investigate occlusal issues or other functional problems. In summary, managing occlusion and protecting implants from overload is an essential part of long-term implant maintenance, and the Corticalization Scale can be a helpful tool for early detection of potentially harmful changes in bone loading.

### 4.7. Assessing Bone Quality and Long-Term Outcomes

Clinicians have several methods at their disposal to assess bone status and quality, including mechanical, imaging, and even biochemical approaches. In the context of dental implant planning and follow-up, imaging methods are among the most practical and informative. For example, CBCT provides a visual 3D assessment of the bone architecture, and conventional CT scans can give Hounsfield Unit (HU) measurements that quantify bone density (for reference, air is approximately −1000 HU, fat is around −50 to −100 HU, water is 0 HU, and cortical bone can be +1000 HU or higher, depending on its density). Another modality, dual-energy X-ray absorptiometry (DXA), measures bone mineral density (BMD) and is commonly used in medicine to diagnose osteoporosis [17,34,35,36]. However, many of these techniques are not routinely feasible in dental implantology. DXA, for instance, is not used for site-specific jawbone assessment, and Hounsfield measurements from medical CT are seldom obtained for dental implant cases due to cost and radiation considerations. In practice, dentists rely on CBCT (and occasionally medical CT when already available) primarily to evaluate bone volume and quality before implant placement.

Our study indicates that baseline bone quality at the time of surgery (implant placement), while important for initial stability, is not the only reliable predictor of marginal bone loss five years later. In other words, just knowing the bone density or quality on the day of implant insertion did not allow us to foresee which implants would eventually experience significant bone loss. This underscores the idea that the long-term success of an implant is influenced by many factors during the healing and post-operative period—such as the functional loading conditions, the patient’s oral hygiene practices, and overall health status—rather than by bone quality alone at baseline.

Crucially, however, we found that by 3 months after functional loading, the situation changes: the early bone response around the implant (as captured by the Corticalization Index on a follow-up radiograph) can indeed predict the likelihood of future bone loss by the 5-year mark. This suggests that the bone’s reaction to the implant under load is a critical indicator of long-term outcome. A favorable early bone response (low CI, indicating retained trabecular structure) is a good sign, whereas an unfavorable response (high CI, indicating rapid corticalization) is a warning sign. With this knowledge, clinicians can use the 3-month post-loading visit not just as a routine check, but as a prognostic evaluation. By identifying patients at higher risk of bone loss early, targeted interventions—such as adjusting the prosthetic load, enhancing maintenance care, or more frequent monitoring—can be implemented to improve the long-term success of the implant.

Limitations of the study were the exclusion of early failures, the lack of multivariable adjustment, the absence of internal and external validation, and the lack of blood tests after the implant insertion.

Despite these limitations, the study offers a solid foundation for further exploration and highlights several clinically relevant trends. Future research is both necessary and encouraged.

## 5. Conclusions

Bone corticalization around dental implants may have a negative impact on bone preservation, depending on its value. The authors created and proposed a new scale. Five-year marginal bone loss can be predicted from a radiograph taken 3 months after functional loading, and evaluating the degree of peri-implant bone corticalization. Thanks to this method, patients will be able to receive new or modified treatment plans that will lead to better implant outcomes.

In conclusion, careful planning and patient selection, rigorous follow-up (with both clinical exams and appropriate imaging), attention to prosthetic design and occlusion, and the use of new tools like the Corticalization Index for early detection of adverse bone changes can all contribute to reducing complications and improving the 5-year success rates of dental implants. The first few months of implant loading emerge as a critical period where the foundation for long-term bone stability is set, and by paying close attention during this period, we can predict and hopefully prevent marginal bone loss around dental implants.

## Figures and Tables

**Figure 1 diagnostics-15-03101-f001:**
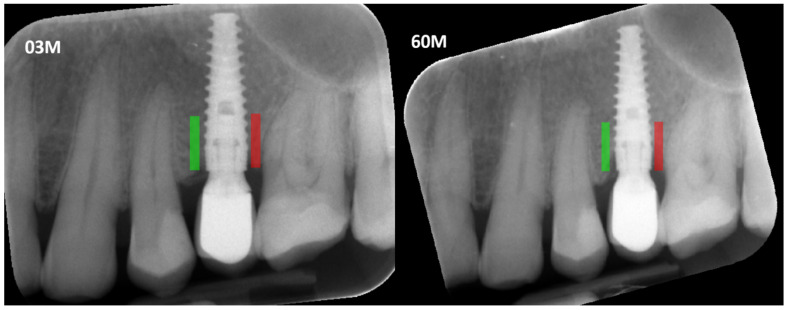
Marking an ROI. ROIs were marked near the implant neck area. Green area—mesial implant neck area; red area—distal implant neck area. Abbreviations: ROI—Region of Interest; 03 M—3 months after functional loading; 60 M—60 months after functional loading.

**Figure 2 diagnostics-15-03101-f002:**
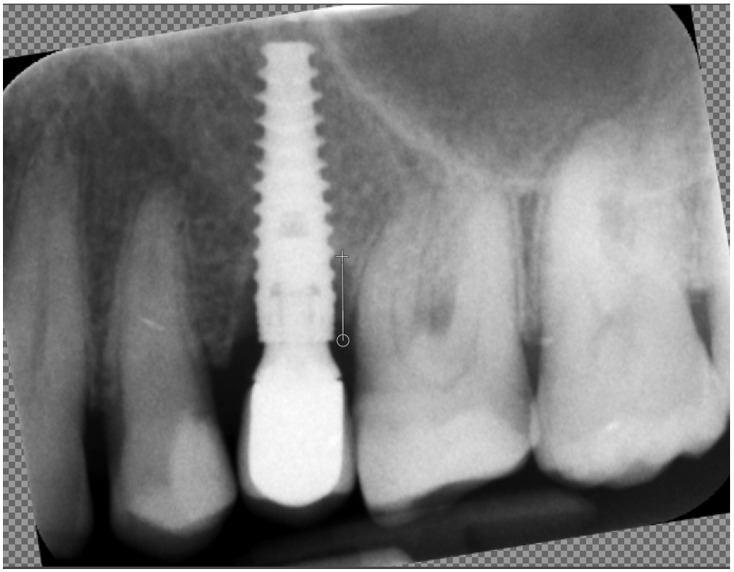
Measuring marginal bone loss on the radiographic images. The white line indicates the implant platform to the bottom of the bone loss cavity.

**Figure 3 diagnostics-15-03101-f003:**
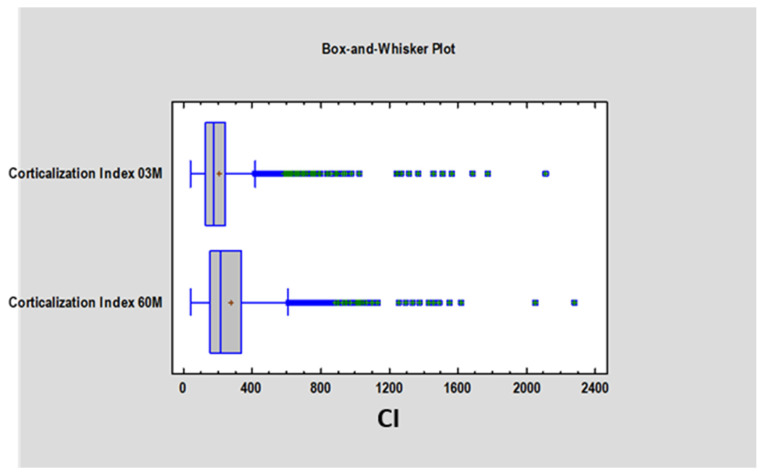
Figure presents the values of the Corticalization Index 3 months after implant, functional loading and the Corticalization Index 60 months after functional loading. Abbreviations: 03 M—3 months after functional loading; 60 M—60 months after functional loading.

**Figure 4 diagnostics-15-03101-f004:**
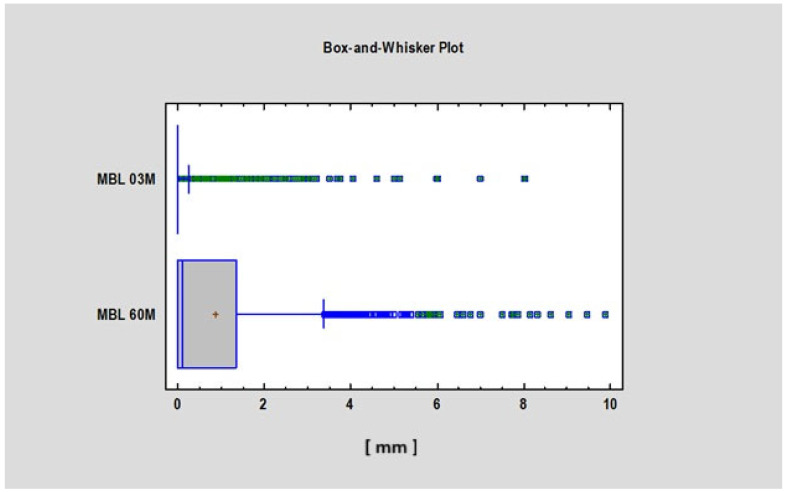
Figure presents the values of MBL 3 and 60 months after implant functional loading. Abbreviations: MBL—marginal bone loss; 03 M—3 months after functional loading; 60 M—60 months after functional loading.

**Figure 5 diagnostics-15-03101-f005:**
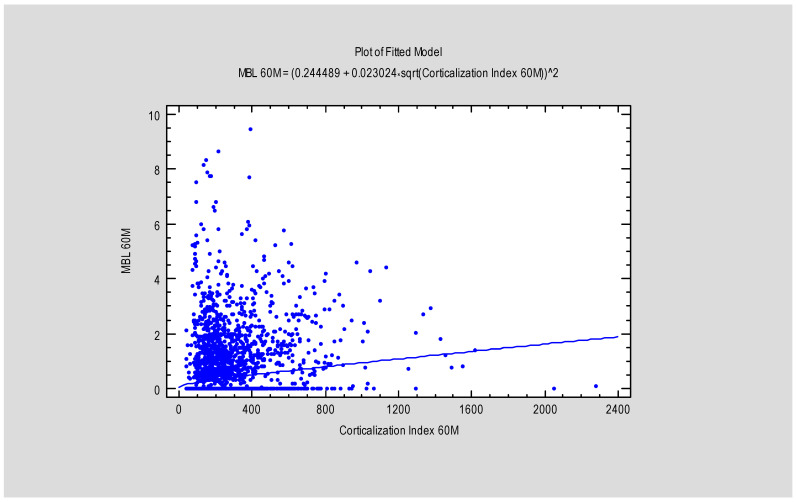
Figure presents the dependence between MBL after 60 months and the Corticalization Index, also after 60 months. Abbreviations: MBL—marginal bone loss; 60 M—60 months of follow-up.

**Figure 6 diagnostics-15-03101-f006:**
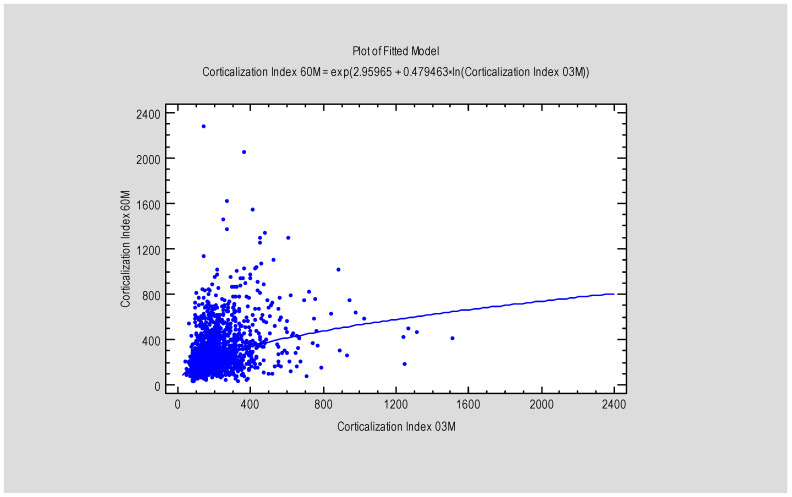
Figure presents the dependence between MBL after 3 months and the Corticalization Index, also after 3 months. Abbreviations: MBL—marginal bone loss; 03 M—3 months of follow-up.

**Figure 7 diagnostics-15-03101-f007:**
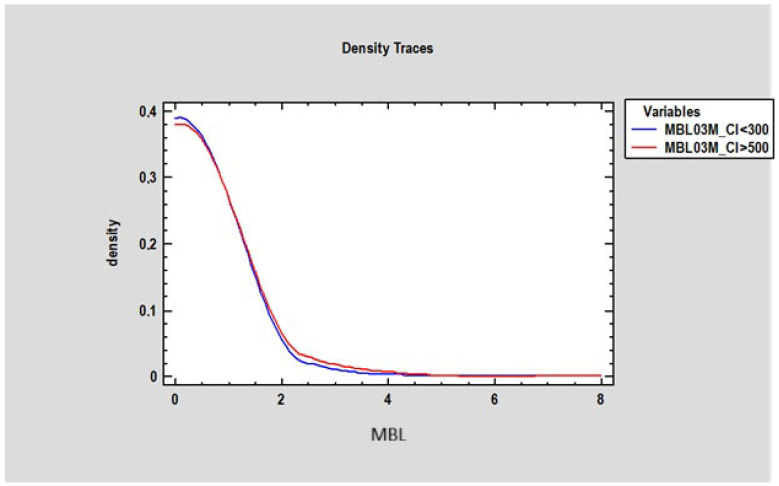
Figure presents changes in marginal bone loss depending on bone corticalization in two groups of patients: the first group of patients, where the Corticalization Index (CI) was lower than 300 (blue line), and the second group with a Corticalization Index higher than 500 (red line) after 3 months of functional loading. Note: Overlapping lines mean the data are equal.

**Figure 8 diagnostics-15-03101-f008:**
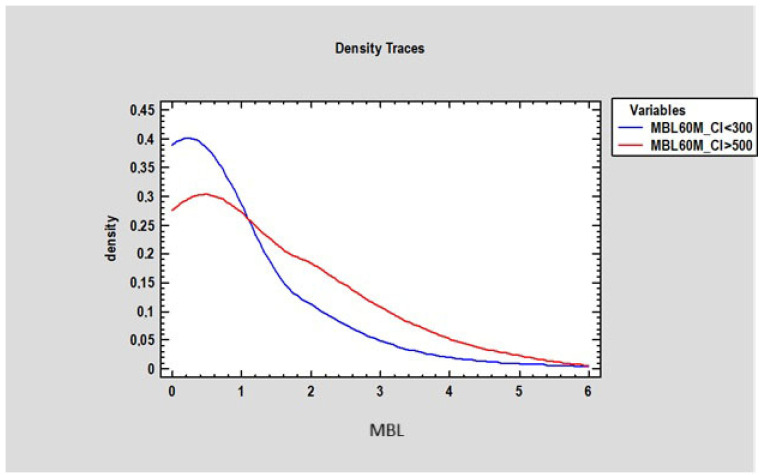
Figure presents changes in marginal bone loss (in millimeters on the horizontal axis) depending on bone density in two groups of patients: the first group of patients where the Corticalization Index was lower than 300 (blue line), and the second group with a Corticalization Index higher than 500 (red line) after 60 months of observation.

**Figure 9 diagnostics-15-03101-f009:**
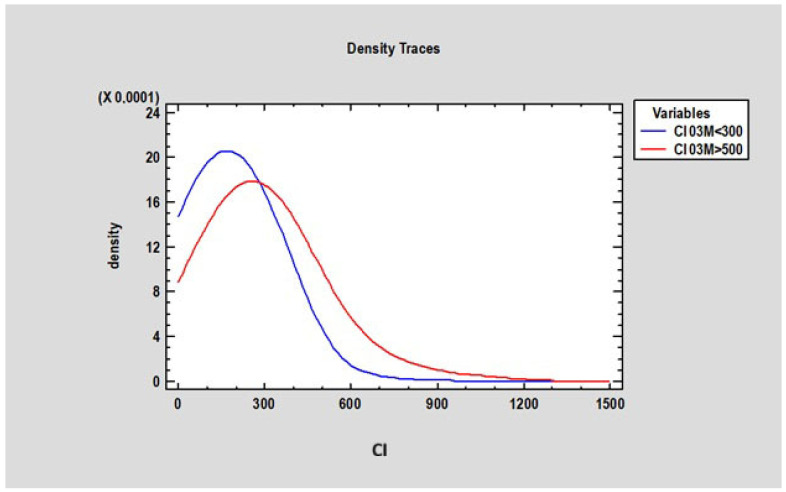
Figure presents changes in the Corticalization Index in two groups of patients: the first group of patients, where the Corticalization Index was lower than 300 (blue line), and the second group, with a Corticalization Index higher than 500 (red line), after 60 months of observation.

**Figure 10 diagnostics-15-03101-f010:**
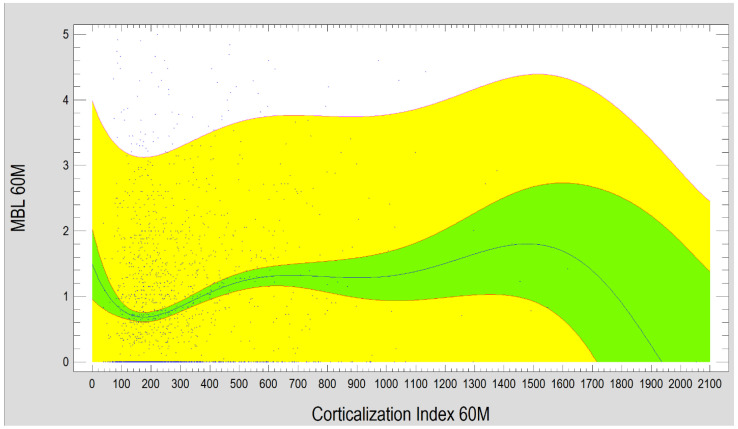
Diagram presents the dependence between MBL and the Corticalization Index after 60 months of observation. Charts show that the higher the Corticalization Index is, the higher the MBL is observed. The critical moment is above 1200 Cortical Index, where MBL is the highest. Abbreviations: MBL—marginal bone loss; 60 M—60 months of observation, yellow—prediction limits; green—confidence limits.

**Table 1 diagnostics-15-03101-t001:** Corticalization Index (CI) and marginal bone loss (MBL)—Summary Table.

Metric	Follow Up	Mean	SD	Notes
CI	0 months (Immediately)	175.02	127,857	-
	3 months	210.70	149.78	Significant vs. 0 M (*p* < 0.05)
	60 months	277.88	198.78	Significant vs. 3 M (*p* < 0.05)
MBL	0 months (Immediately)	0.00 mm	0.04 mm	-
	3 months	0.24 mm	0.90 mm	Significant vs. 0 M (*p* < 0.05)
	60 months	0.86 mm	1.29 mm	Significant vs. 3 M (*p* < 0.05)

Table presents the values of the Corticalization Index and marginal bone loss over time. Abbreviations: CI—Corticalization Index; MBL—marginal bone loss; SD—standard deviation. *p* < 0.05 was statistically significant.

**Table 2 diagnostics-15-03101-t002:** Correlation summary (CI and MBL).

Correlation	Variables	Mean ± SD	(*p*-Value)
CI 60 M ↔ MBL 60 M	CI 60 M: 277.88 ± 198.78 MBL 60 M: 0.86 ± 1.29	CI 60 M: 277.88 ± 198.78 MBL 60 M: 0.86 ± 1.29	Significant (*p* < 0.05)
CI 60 M ↔ CI 03 M	CI 60 M: 277.88 ± 198.78 CI 03 M: 210.70 ± 149.78	CI 60 M: 277.88 ± 198.78 CI 03 M: 210.70 ± 149.78	Significant (*p* < 0.05)

Table presents the correlation between Corticalization Index and marginal bone loss. Abbreviations: CI—Corticalization Index; MBL—marginal bone loss; SD—standard deviation; 03 M—3 months of observation; 60 M—60 months of observation.

## Data Availability

The original contributions presented in this study are included in the article. Further inquiries can be directed to the corresponding author.

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
