# Peer review of "Proposition of a New Scale for Marginal Bone Loss Prediction Around Dental Implants—A 5-Year Follow-Up of Functional Loaded Implants"

_diagnostics, 2025, doi:10.3390/diagnostics15243101_

Round 1

Reviewer 1 Report

Comments and Suggestions for Authors

The paper targets a meaningful clinical task: forecasting peri-implant marginal bone loss using texture metrics from routine intra-oral radiographs. Its main idea—a Corticalization Index (CI)–based risk scale derived from standard images—is novel and potentially inexpensive. Yet numerous methodological and reporting gaps undermine confidence in the results and their clinical applicability.

  1. Implants, not patients, are the analysis unit, inviting clustering/non-independence that isn’t modeled.
  2. Failing to account for multiple implants per patient likely yields anti-conservative p-values and overconfident inferences.
  3. Removing implants lost before three months introduces survivorship bias, potentially diluting associations with failure-prone cases.
  4. Highly selective inclusion criteria favor ideal conditions, limiting generalizability to everyday mixed-risk populations.
  5. A single surgeon and uniform protocol aid internal consistency but constrain applicability across clinicians and centers.
  6. Key implant characteristics (brand, surface, connection, platform switching, diameter, length) are unreported, obscuring confounding.
  7. Site variables (maxilla/mandible, anterior/posterior, bone class) are missing, hindering interpretation of heterogeneity.
  8. Important patient confounders (smoking intensity, bruxism, parafunction, glycemic control) aren’t analyzed in adjusted models.
  9. No multivariable or mixed-effects analyses are used to handle confounding or repeated measures across time.
  10. Although the 0, 3, and 60-month visits are reasonable, attrition, missing-data handling, and denominators per visit are not detailed.
  11. Five-year follow-up completeness is unclear, impeding assessment of selection bias over time.
  12. Measuring MBL on 2D radiographs is vulnerable to projection/magnification errors despite positioners.
  13. The “long-axis correction” lacks reproducible detail, and no calibration standard is reported to normalize scale.
  14. Inter- and intra-observer reliability for ROI placement and MBL readings is absent—critical for imaging research.
  15. The texture pipeline depends on MaZda 4.6 and custom feature composition without shared code, protocol, or phantom calibration.
  16. Digital optical density isn’t a linear proxy for attenuation without calibration, challenging the CI’s physical validity.
  17. The CI formula (LngREmph × mean optical density / DiffEntr) is given with little biological rationale and no sensitivity testing to feature choices.
  18. Equations include typos/ambiguous notation, casting doubt on exact implementation.
  19. CI cutoffs (<300, >500, >1200) appear post hoc and lack prespecification or cross-validation.
  20. No ROC/AUC, calibration, or decision-curve analyses are provided to justify thresholds or clinical utility.
  21. Transformations (square-root, log) are shown without fit indices, residual checks, or R².
  22. Repeated “p<0.5” typographical errors erode statistical credibility and editorial quality.
  23. Small early MBL differences (e.g., 0.24 mm at three months) achieve significance, but clinical importance isn’t discussed.
  24. CI shows high variance, implying overlap between risk strata and potential individual-level misclassification.
  25. The scale is advocated for decision-making without prospective validation or proof of outcome benefit.
  26. Texture likely depends on device and site (exposure settings, sensors), yet no external validation is presented.
  27. The claim that 2D radiographs outperform CBCT for texture analysis is asserted, not quantitatively demonstrated.
  28. Ethics approval is noted, but informed consent, demographics, and baseline patient characteristics are not reported.
  29. Follow-up cadence and adherence lack detail, and reasons for exclusions are not fully itemized.
  30. Excluding augmented sites eases interpretation but omits a large portion of real-world implant cases.
  31. Prosthetic determinants (emergence profile, connection/retention type, cantilevers, contacts) are acknowledged yet unmeasured.
  32. “Prediction” language risks implying causality that the observational design and limited confounder control cannot support.
  33. Figures may duplicate information; axes/units are inconsistently labeled; raw summary tables are sparse.
  34. Numerous grammatical errors and fragmentary sentences impede clarity and may conceal methodological ambiguities.
  35. Nonetheless, the core signal—greater early corticalization associating with increased later MBL—is biologically plausible.
  36. Future work should involve preregistration, transparent protocols, and mixed-effects models with patient- and site-level random effects.
  37. Reporting ought to include patient counts, implants per patient, and stratification by site and implant system.
  38. Measurement reliability should be quantified via ICCs and Bland–Altman analyses for ROI and MBL assessments.
  39. Step-wedge or phantom calibration is needed to stabilize optical-density-based features across sessions/devices.
  40. Threshold derivation should use ROC optimization with bootstrapped CIs and internal cross-validation.

Comments on the Quality of English Language

Improvement required 

Author Response

Reviewer letter in attachement.

Reviewer 2 Report

Comments and Suggestions for Authors

Dear authors,

Congratulations on a job well done.

The abstract of the manuscript is very comprehensibly written and comprehensive.
The introduction provides enough information to guide readers to the essence of the article. At the end, you have clearly stated the purpose of the study.
The selection through the inclusion and exclusion criteria is reliable.
It seems that you have used appropriate statistical analysis in processing the data, and when visualizing them, you have used appropriate and understandable diagrams for the readers.
The discussion is well presented.
The conclusion is appropriate for the results obtained.

To improve the manuscript, I suggest that you insert the following clarifications:

1. Write the date of the committee's decision for the following sentence: "This study was conducted with the approval of the institutional bioethics committee (approval numbers: RNN/485/11/KB )".

2. Write when the study was conducted (from when to when).

3. Write where the study was conducted - clinic, hospital, city and country.

4. At the end of the discussion, write what the limitations of the study are.

Thank you!

Author Response

Dear Reviewer,

Thank you for the positive evaluation of our manuscript; we truly appreciate your remarks and suggestions. All of your comments have been addressed. We hope that the revisions we have made will lead to the acceptance and publication of our work.

  1. Write the date of the committee's decision for the following sentence: "This study was conducted with the approval of the institutional bioethics committee (approval numbers: RNN/485/11/KB )". - corrected
  2. Write when the study was conducted (from when to when). – it was retrospective study and the material was collected for many years.
  3. Write where the study was conducted - clinic, hospital, city and country.- Clinic, Poland, Łódź
  4. At the end of the discussion, write what the limitations of the study are.- added

Reviewer 3 Report

Comments and Suggestions for Authors

General Comments:

This manuscript presents an innovative approach for predicting marginal bone loss (MBL) around dental implants using a novel Corticalization Scale based on the Corticalization Index (CI). The study addresses an important clinical issue, as MBL significantly affects long-term implant success.
A notable strength is the use of widely available two-dimensional intra-oral radiographs to calculate the CI, enhancing the method’s clinical applicability and accessibility. The authors analyzed a large dataset and demonstrated a clear correlation between higher CI values after 3 months of functional loading and increased MBL after 60 months.
Overall, this is a relevant and promising study; however, a few points require clarification to improve internal consistency and reproducibility.

Major comments:

Inconsistency in CI thresholds for MBL risk classification
The manuscript states that CI > 500 at 3 months indicates a higher risk of MBL at 5 years, while Figure 12 (“Corticalization Scale and the risk of MBL after 60 months”) defines High risk as CI > 1200. Please clarify the rationale for setting 1200 as the high-risk threshold.  Consistency between text and figure should be ensured.

Minor Comments

  1. Figure 8 legend: Correct the misspelling of “functional” in the third line.
  2. Figures 3, 4, 7, 8, and 9: Clearly indicate the x-axis labels and units of measurement to facilitate interpretation.

Author Response

Dear Reviewer,

We sincerely thank you for your positive assessment of our manuscript and greatly appreciate the comments you provided. All of your suggestions have been fully incorporated. We hope that the implemented revisions will meet your expectations and result in the acceptance and publication of our work.

Major comments

Inconsistency in CI thresholds for MBL risk classification
The manuscript states that CI > 500 at 3 months indicates a higher risk of MBL at 5 years, while Figure 12 (“Corticalization Scale and the risk of MBL after 60 months”) defines High risk as CI > 1200. Please clarify the rationale for setting 1200 as the high-risk threshold.  Consistency between text and figure should be ensured.

We hope that corrected manuscript describe it better.

Minor Comments

  1. Figure 8 legend: Correct the misspelling of “functional” in the third line. - corrected
  2. Figures 3, 4, 7, 8, and 9: Clearly indicate the x-axis labels and units of measurement to facilitate interpretation. – corrected

Round 2

Reviewer 1 Report

Comments and Suggestions for Authors
  • Please clearly state what is genuinely new in this manuscript compared with your prior work on Corticalization and the Corticalization Index (CI). It appears that the main addition is the introduction of CI-based risk categories (<300, >300, >1200), but it is unclear whether the dataset itself is new or a re-analysis of previously published cases, and what is added beyond your 2022 and 2024 studies?

  • The choice of CI cut-offs (<300, >300, >1200) for low/medium/high risk of marginal bone loss (MBL) requires a transparent and rigorous justification. 

  • The flow from 1270 implants (820 patients; 2540 ROIs) to 1026 included implants needs clearer description. Please specify the number and reasons for all exclusions (especially early failures within 3 months),

  • Because multiple implants can belong to the same patient (1026 implants vs 820 patients), independence of observations is doubtful, and no multilevel or clustered models are shown.

  • The description of t-tests, W-tests, ANOVA/Kruskal–Wallis, and regression is too general. For key models (e.g., MBL60M vs CI60M; CI60M vs CI3M), please provide full model summaries, including which tests were used for which comparisons and effect estimates with confidence intervals. If CI is proposed as a predictive tool, add prognostic performance measures (e.g., discrimination, calibration), not only p-values.

  • You report mean MBL of 0.86 ± 1.29 mm at 60 months and an association with higher CI, but the clinical significance remains unclear. Please describe the full MBL distribution (including outliers).

  • Given that CI is central to your conclusions, please provide reliability statistics (e.g., ICCs, Bland–Altman analyses) or, at minimum, detailed procedures to ensure reproducibility.

  • Expand the rationale for why baseline bone status is not predictive, particularly when proposing CI as a superior radiographic measure.

  • The limitations section is too narrow relative to the strength of the claims. explicitly acknowledge the retrospective design, exclusion of early failures, lack of multivariable adjustment, absence of internal/external validation, etc, and explain how these limit the strength and generalizability of your conclusions.

  • The manuscript requires substantial language and structural editing. Numerous grammatical errors (e.g., “method, than can may predict bone loss”; “Marginal Bone Loos”; “has become widespread technique”), inconsistent terminology , and unclear figure/table captions currently obscure the scientific message. 

Comments on the Quality of English Language

Improvement required 

Author Response

Answers and letter to the Reviewer in attachment. 
